# The Complete Mitochondrial Genome of *Pilumnopeus Makianus* (Brachyura: Pilumnidae), Novel Gene Rearrangements, and Phylogenetic Relationships of Brachyura

**DOI:** 10.3390/genes13111943

**Published:** 2022-10-25

**Authors:** Xinbing Duan, Xiangli Dong, Jiji Li, Jiayin Lü, Baoying Guo, Kaida Xu, Yingying Ye

**Affiliations:** 1National Engineering Research Center for Marine Aquaculture, Zhejiang Ocean University, Zhoushan 316022, China; 2Key Laboratory of Sustainable Utilization of Technology Research for Fishery Resource of Zhejiang Province, Marine Fishery Institute of Zhejiang Province, Zhoushan 316021, China

**Keywords:** *Pilumnopeus makianus*, pilumnidae, mitogenome, phylogenetic analysis

## Abstract

*Pilumnopeus makianus* is a crab that belongs to Pilumnidae, Brachyura. Although many recent studies have focused on the phylogeny of Brachyura, the internal relationships in this clade are far from settled. In this study, the complete mitogenome of *P. makianus* was sequenced and annotated for the first time. The length of the mitogenome is 15,863 bp, and includes 13 protein-coding genes (PCGs), 22 transfer RNA genes (tRNA), and 2 ribosomal RNA genes (rRNA). The mitogenome exhibits a high AT content (72.26%), with a negative AT-skew (−0.01) and a GC-skew (−0.256). In the mitogenome of *P. makianus*, all the tRNA genes are folded into the typical cloverleaf secondary structure, except trnS1 (TCT). A comparison with the ancestors of Brachyura reveals that gene rearrangement occurred in *P. makianus*. In addition, phylogenetic analyses based on thirteen PCGs indicated that *P. makianus*, *Pilumnus vespertilio*, and *Echinoecus nipponicus* clustered into a well-supported clade that supports the monophyly of the family Pilumnidae. These findings enabled a better understanding of phylogenetic relationships within Brachyura.

## 1. Introduction

The “true” crabs in the infraorder Brachyura [1] belong to the Arthropoda, Malacostraca, Decapoda. It is one of the most diverse groups among the decapods [2]. The infraorder Brachyura contains more than 7250 known species in 104 families [3], which are widely distributed from shallow coral reefs to hydrothermal vents in the ocean, as well as freshwater and terrestrial habitats [4]. The current classification of Brachyura was established by Guinot [5] based on gonopore position morphology, and is currently divided into three sections (i.e., Podotremata, Heterotremata, and Thoracotremata) [5]. A high incidence of derived characteristics has given rise to many controversies concerning brachyuran phylogenetic relationships. Pinnotheroidea, Hymenosomatoidea, and Hexapodidae were originally assigned to the Thoracotremata [5]. However, Guinot and Forges [6] argued that these three superfamilies should be assigned to the Heterotremata based on the gonopore location and ultrastructure of the male reproductive system. Von Sternberg et al. [7] re-established a phylogenetic relationship between Potamoidea and other crabs by considering morphological features. They assigned Potamoidea to Heterotremata and further identified a sister-group relationship with Thoracotremata, thereby embedding Potamoidea within Thoracotremata [7].

The superfamily Pilumnidea is widely distributed along the western Pacific coast, and consists of three families (i.e., Pilumnidae, Galenidae, and Tanaochelidae). At present, only two complete mitogenomes of Pilumnidea have been published in the NCBI database (i.e., *P**. vespertilio* (MF457402) and *E. nipponicus* (NC_039618)). The crab *P. makianus* belongs to Pilumnidae, which inhabits the rocky and muddy shores of tropical and subtropical seas. Some previous studies classified the Pilumnidae as a subfamily of Xanthidae [8]. This classification was challenged by Guinot [5], who elevated the Pilumninae to Pilumnidae due to their exhibition of characteristics more advanced than those found in other Xanthoidea families. Guinot [5] also elevated the Xanthidae to Xanthoidea based on the position of the male genital foramen. Subsequently, with the rise of comparative morphology research, the classification of Pilumnidea has been revised again. Due to the special genital characteristics of males (i.e., a slender and curved first abdominal limb, and a prominent, short, “S”-shaped second abdominal limb), Pilumnidae was elevated into Pilumnidea [9,10]. However, the morphology-based phylogenetic analysis presents problems [11,12], and the whole process of identification by morphology is still controversial. The analysis of the mitogenomic sequence data might provide a useful approach for identifying new brachyuran species and analyzing brachyuran phylogenetics.

The mitochondrial genome in Brachyura is usually a closed loop constituted of 13 protein-coding genes (PCGs), 22 transfer RNA genes (tRNAs), 2 ribosomal RNA genes (12S and 16S), and an AT-rich region (also called a control region, or *CR*) [13]. In phylogenetic studies, PCGs are commonly used to construct phylogenetic trees. The protein-coding genes include three cytochrome oxidase subunits (*COX1-3*), one cytochrome dehydrogenase subunit (*CYTB*), two ATP synthase subunits (*ATP6* and *ATP8*), and seven NADH dehydrogenase subunits (*ND1-6* and *ND4L*) [14]. Due to the long evolutionary history of symbiosis between mitochondria and eukaryotes, their genomes contain key messages reflecting their evolution as species [15]. Their mitochondrial genomes were characterized by fast evolution, maternal genetics, simple molecular structure, and relatively easy acquisition [16]. In general, mitochondrial rearrangement is common in invertebrates, including Cephalopoda [17], Bivalvia [18], and Brachyura [19]. Therefore, a complete mitogenome would be useful for analyzing gene rearrangements and identifying evolutionary relationships.

In this study, we sequenced the complete mitogenome of *P. makianus*. We first obtained the mitogenome map of *P. makianus* to determine the location of genes and confirm its nucleotide composition. The gene rearrangement and phylogeny were analyzed to improve the level of classification and phylogenetic within Brachyura.

## 2. Materials and Methods

### 2.1. Sample Collection and DNA Extraction

We collected one specimen of *P. makianus* from Changzhi Island, Zhoushan, Zhejiang Province, China (122°14′ N, 29°97′ E) on 27 September 2020. We identified the specimen by morphology, then excised the epaxial musculature and saved tissue in absolute ethyl alcohol for the purpose of DNA extraction. The total genomic DNA was extracted by the salt-extraction procedure with a minor modification [20], then stored in 1 × TAE buffer at 4 °C after being extracted immediately. We then used 1.5% agarose gel electrophoresis to identify the extracted DNA and then stored it at −20 °C.

### 2.2. Sequence Assembly, Annotation, and Analysis

The *P. makianus* mitogenome was sequenced by Origin gene Co. Ltd., Shanghai, China, on the Illumina HiSeq X Ten platform (Illumina, CA, USA). Each library generated about 10 Gb of raw data with inserts of 300–500 bp size from genomic DNA. The reads and adapters of low-quality, sequences with high “N” ratios and fragments with lengths less than 25 bp were moved away. We aggregated clean reads using NOVOPlasty software [21]. To validate the sequences, we compared the aggregated mitochondrial genes to the other Pilumnidea species and checked the *COX1* sequence in NCBI BLAST (https://blast.ncbi.nlm.nih.gov/Blast.cgi (accessed on 1 March 2022)) to verify the mitogenomic sequences [22]. We determined aberrant start and stop codons by comparison with similar codons in other invertebrate species. We then used the de novo assembly program to rebuild the reads. The software Sequin 16.0 (https://www.ncbi.nlm.nih.gov/sra (accessed on 1 March 2022)) was used to annotate the complete mtDNA, and the MITOS web server [22] was used to verify the correctness of transfer RNA genes and their secondary structures. Then CGView [23] was used to obtain the mitogenome map of *P. makianus* (Figure 1). We analyzed base composition by DAMBE [24] and obtained relative synonymous codon usage (RSCU) using MEGA XI [25]. To calculate strand asymmetry, we used the following formulae: AT-skew = (A − T) / (A + T); and GC-skew = (G − C)/(G + C) [26].

### 2.3. Phylogenetic Analysis

We downloaded 76 complete mitogenome sequences, including outgroups *Pagurus nigrofascia* and *Pagurus gracilipes*, from NCBI (Appendix A). We determined the phylogenetic relationship of Brachyura using the mitogenome sequences of the 76 species and *P. makianus*. Thirteen PCGs from the mitogenome sequences of all species were extracted from the GenBank files using DAMBE [24]. We then aligned the genes in MEGA XI [25] and constructed a single alignment file that connected all sequences, and then carried out a format conversion to create nexus format files and paste files that we used for our phylogenetic analyses. We used DAMBE [24] to detect the saturation of protein-coding genes and to remove supersaturated genes. The maximum likelihood (ML) and Bayesian inference (BI) were employed to construct a phylogenetic tree. Based on the Bayesian information criterion (BIC), we selected the best-fit model GTR + F + R6 for each subarea, and then used IQ-TREE [27] with 1000 replicates for ML analysis. For BI analysis, after applying the Akaike information criterion (AIC) in MrModelTest 2.3 [28], we confirmed best-fit evolutionary models (GTR + I + G) in MrMTgui [29]. The PAUP, ModelTest, and MrModelTest were associated through MrMTgui across platforms. We then ran the BI analysis using 3×10^6^ Markov Chain Monte Carlo (MCMC) sampling and estimated the posterior distribution. We sampled every 1000 generations and discarded burn-in for 25% of the generations. To guarantee stationarity, we set the average standard deviation of split frequencies below 0.01. Finally, we used Figure Tree v1.4.3 software [30] to visualize the phylogenetic trees. In this research, we used Baeza’s [31] workflow to guide our detailed analysis, which includes the de novo assembled, annotated, manually curated, and characterized of mitochondrial genomes.

## 3. Results and Discussion

### 3.1. Base Structure and Composition of the Mitogenome

The complete mitogenome sequence of *P. makianus* is 15,863 bp in length. It has been uploaded to GenBank under accession number OM461360, and the raw reads were deposited in GenBank with the SRA number SRR21765684. The mitogenome content of *P. makianus* is a closed-circular molecule that includes 13 PCGs, 22 tRNAs, 2 rRNAs (*rrnl* and *rrns*), and a small non-coding region (Table 1). 

The mitogenome sequence of *P. makianus* shares common features with other Brachyura whose genes have been identified and published. A total of four PCGs (*ND5, ND4, ND4L*, and *ND1*), eight tRNAs (*tRNA-His, Phe, Pro, Tyr, Leu, Val, Gln, Cys*), and two rRNAs are located in the light (L–) strand; the remainder of the 37 genes are located in the heavy (H–) strand (Table 1). The nucleotide composition of the *P. makianus* complete mitogenome is as follows: 35.76% A, 36.5% T, 10.32% G, and 17.42% C (Table 2). In addition, the *P. makianus* mitogenome has a high AT bias (72.26%). While its GC-skew and AT-skew values are negative, being −0.256 and −0.01, respectively, indicating that Cs and Ts exceed Gs and As (Table 2). In short, for *P. makianus,* the nucleotide composition, the GC-skew and AT-skew of the total mitogenomes, and the gene sequence length are all similar to the other Pilumnidea species (Table 3).

Brachyura mitochondrial genomes are compact in structure, and include some overlaps between adjacent coding genes. However, gene spacers are also present. The *P. makianus* mitogenome possessed 21 intergenic spacers and seven overlapping regions. The seven overlaps range from 1 to 56 bp, including three representative overlaps in protein-coding genes (1 bp between *ATP6* and *COX3*, 5 bp between *ND4* and *ND4L*, 1 bp between *ND6* and *CYTB*), which are typically found in other crabs.

### 3.2. PCGs and Codon Usage

The total length of PCGs in the *P. makianus* mitogenome is 11,066 bp, including seven NADH dehydrogenases (*ND1-6* and *ND4L*), two ATPases (*ATP6* and *ATP8*), three cytochrome c oxidases (*COX1-3*), and one cytochrome b (*CYTB*). All thirteen PCGs begin with the start codon ATN (ATA, ATG, ATC, and ATT). The majority of the 13 PCGs end with TAA or TAG, while the stop codon of *CYTB* is a single T (Table 1). Incomplete stop codons are a common phenomenon in the mitochondrial genes of vertebrates and invertebrates [32]. Figure 2 demonstrates the relative synonymous codon usage (RSCU) and the amino acid compositions of the *P. makianus* mitochondrial genome. The most frequently occurring codon is UUA-Leu (2.23) (Figure 2a), and the most common amino acids are Ile (7.0%), Asn (8.0%), Phe (8.9%), and Ile (11.0%). The least common amino acids are Glu (1.3%), Arg (1.3%), Asp (2%), and Cys (2%) (Figure 2b). 

### 3.3. Transfer and Ribosomal RNAs

In common with other crabs, the mitogenome of *P. makianus* contains 22 tRNA genes. The length of these ranges from 63 bp (*trnC*, *trnA*) to 72 bp (*trnV*) of nucleotides (Table 1). The overall A + T content of tRNA genes is 73.11%; in addition, they exhibit a positive AT skew (0.023) and GC skew (0.156) (Table 2). As *trnS1* lacks the dihydrouridine (DHU) arm, only *trnS1* (TCT) cannot be folded to a representative cloverleaf secondary structure (Figure 3). This phenomenon is common in metazoans [33].

In the *P. makianus* genome, *rrnS* and *rrnL* have lengths of 831 bp and 1355 bp and are separated by *trnV*. The 12S rRNA gene is followed by *trnV*, and the 16S rRNA gene is situated between *trnL2* (TAA) and *trnV*. The A + T content of rRNAs is 76.2%, AT-skew (0.030) and GC-skew (0.322) are both positive (Table 2), indicating that As and Gs exceed Ts and Cs.

### 3.4. Gene Rearrangement

In general, the mitochondrial gene orders (MGOs) of invertebrates exhibit gene rearrangements on different scales [34,35]. This phenomenon has been demonstrated in the MGOs of *P. makianus* by means of comparison with the ancestor of Brachyura and subsequent analysis. To identify gene rearrangement, previous researchers have used four different models: the tandem duplication-random loss model (TDRL) [36], which is the most commonly used means of explaining the mechanism of mitochondrial genome rearrangement, the tandem duplication-nonrandom loss model (TDNL) [37], recombination [38], and tRNA miss-priming model [39]. 

We compared the MGOs of Pilumnidae and other Heterotremata species with ancestral Brachyura (Figure 4). We found the MGOs of the families Leucosiidae, Matutidae, Portunidae, and Oregoniidae were identical to ancestral Brachyura. However, the families Pilumnidae, Xanthidae, and Majidae have undergone gene rearrangements. Compared with the ancestral Brachyura, *P. makianus* exhibited translocation and duplication-random loss (Figure 4). In *P. makianus* mitogenome, *trnL2* has moved to a position between *trnL1* and *16S* and changed from heavy (H–) strand to light (L–) strand, resulting in a new gene order *trnL1-trnL2-16S-trnV-12S-ND1*, which can be explained by the TDRL model (Figure 4). In most families, MGOs of individual species within the family were largely consistent [40,41]. However, the MGOs of the three Pilumnidae species exhibited considerable differences. For instance, compared with *P. makianus*, the *trnL1* and *trnL2* were inverted in *P. vespertilio*; and in *E. nipponicus*, the *ND1* and *trnL1* moved to the back of *CR* (Figure 4). Within Pilumnidae, Chen et al. [42] have reported the study of *Heteropanope glabra*, and we found the MGOs of *H. glabra* studied by Chen et al. [42] are the same as those of *P. makianus*.

Mitochondrial gene orders provide compelling phylogenetic information. Previous studies suggested that Pilumnidae and Xanthidae were closely related and might even belong to the same family [8]. In this study, by assessing the MGOs of Heterotremata, we found that gene rearrangement had taken place in both Pilumnidae and Xanthidae, and the MGOs of other species within Heterotremata were consistent with ancestral Brachyura, except for the more distant Majidae families. In addition, we found the families Pilumnidae and Xanthidae have undergone different rearrangements, which provided key evidence of their evolutionary history. The difference in the MGOs of Xanthidae [43] and Pilumnidae also provided valuable independent support for the separation of Pilumnidae from Xanthidae, but the low number of published Pilumnidae mitochondrial studies made it difficult to explore the ancestral sequence of this family.

### 3.5. Phylogenetic Relationships

In this study, we used the sequences of 13 PCGs in mitochondrial genomes to establish a phylogenetic relationship of Brachyura. We analyzed *P. makianus* and 74 other brachyuran species, with *P. nigrofascia* and *P**. gracilipes* as outgroups (Appendix A). Numbers above the branches indicate posterior probabilities from BI and bootstrap percentages from ML, respectively. For all species, ML and BI methods generated the same topological structure for each dataset, and the BI tree was demonstrated for high supporting values on the whole. Support values obtained through ML and BI methods were shown on the BI tree, and *P. makianus* was marked with a red dot. It can be observed that *P. vespertilio* and *E. nipponicus* form a sister clade with high support values (BI posterior probabilities PP = 1, ML bootstrap BP = 92%). *P. makianus*, along with those two species, formed a Pilumnidae group with high support values, with maximum values for both BI posterior probabilities and ML bootstrap percentages (Figure 5).

There are 20 families in this phylogeny. Heterotremata was divided into two groups (Majoidea + ((Pilumnoidea+Xanthoidea) + ((Leucosioidea + Calappoidea) + (Cancroidea + Portunoidea)))) + (Potamoidea + Gecarcinucoidea). The species in Majoidea are one of the oldest lineages in brachyuran crabs and are deemed to be the nearest branch to the base of a brachyuran tree, based on the spermatozoal ultrastructure [44]. Our phylogenetic tree confirmed this. Pilumnidae was clustered with Xanthidae and Panopeidae, and formed a sisterhood with another group which includes Portunidae, Matutidae, Leucosiidae, and Cancridae [45]. Previously, Pilumnidae were considered a subfamily of Xanthidae; however, Tsang et al. [11] constructed a Brachyura molecular phylogeny based on six nucleoprotein coding genes and two rRNA genes and verified the Pilumnidea species formed a separate branch, independent of Xanthidea. Our mitochondrial genome confirmed the reliability of this result and further explained the separation of Pilumnidae from Xanthidae, so that the former is now considered a separate family. The specific classification status of *P. makianus* has also been verified from the perspective of morphology and molecular biology [46]. 

The superfamilies Gecarcinucoidea and Potamoidea are freshwater crabs that belong to Heterotremata, but they were found to have a close relationship with Thoracotremata. Von Sternberg and Cumberlidge [47] suggested that most families of freshwater crabs, including Potamidae and Gecarcinucidae, should be placed in the Thoracotremata. The Gecarcinucoidea, Potamoidea, and Thoracotremata clustered into a branch, supporting the conclusion that Heterotremata is polyphyletic. Brosing et al. [48] confirmed the polyphyletism of Heterotremata through morphological identification. Ahyong et al. [49] used a partial sequence of 18s rDNA to reconstruct the phylogeny of Brachyura, and found that the two representative classes of Thoracotremata were completely nested in Heterotremata, thus confirming the paraphyly of Heterotremata. This conclusion was in line with the findings of Ahyong et al. [49] and Bracken et al. [48]. Our tree demonstrates that Thoracotremata is divided into Grapsoidea and Ocypodoidea, and Grapsoidea is further divided into three clades (i.e., (Grapsidea + Varunidae + (Gecarcinidae + Sesarmidae))) and the Ocypodoidea was divided into two clades (i.e., ((Mictyridae + Macrophthalmidae) + (Dotillidae + Ocypodoidae))) (Figure 5). These findings are also consistent with those of previous studies [48,49,50]. Chen et al. [51] found that Gecarcinucoidea and Potamoidea are more connected with thoracotreme crabs, and Thoracotremata is nested within Heterotremata clades. These results are also in accordance with our findings.

## 4. Conclusions

In this study, the complete mitogenome of *P. makianus* was sequenced and described for the first time. The 15863 bp mitogenome of *P. makianus* includes 37 genes and a AT-rich region, in common with the metazoan mitogenome. The genome composition exhibits a high A + T biased (72.26%) and demonstrates a negative AT-skew (–0.01) and GC-skew (–0.256). Compared with the ancestor of Brachyura, there has been a genetic rearrangement in *P. makianus* mitogenomes, which can be explained by the TDRL model. The phylogenetic tree was structured by 77 species, and was divided into two branches. The *P.*
*makianus* exhibited the closest relationship with *P. vespertilio* and *E. nipponicus*, and these three species formed the Pilumnidae cluster. We also explained the reason why genetic scientists separated Pilumnidae from Xanthidae. Our research results enable a better understanding of taxonomy and phylogenetic relationships within Brachyura. 

## Figures and Tables

**Figure 1 genes-13-01943-f001:**
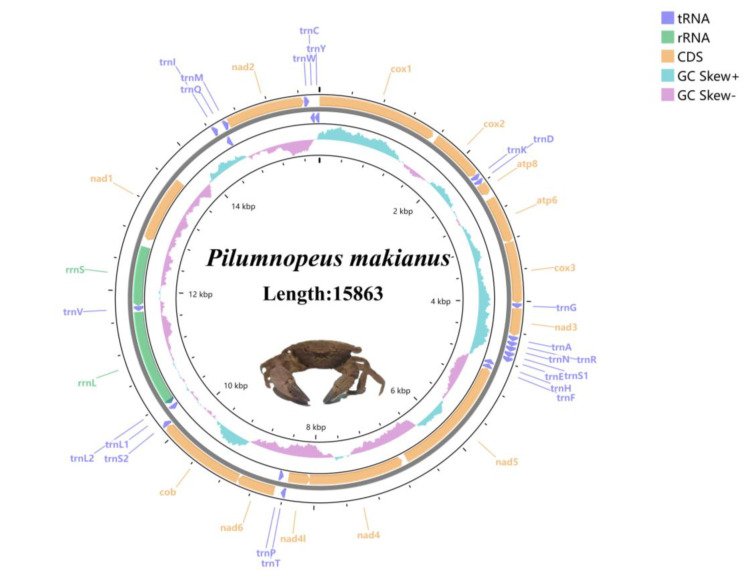
Mitogenome map of the *P. makianus*. Lavender arrows represent tRNA, green arrows represent rRNA, and yellow arrows represent CDS. Blue represents GC skew + and pink represents GC skew.

**Figure 2 genes-13-01943-f002:**
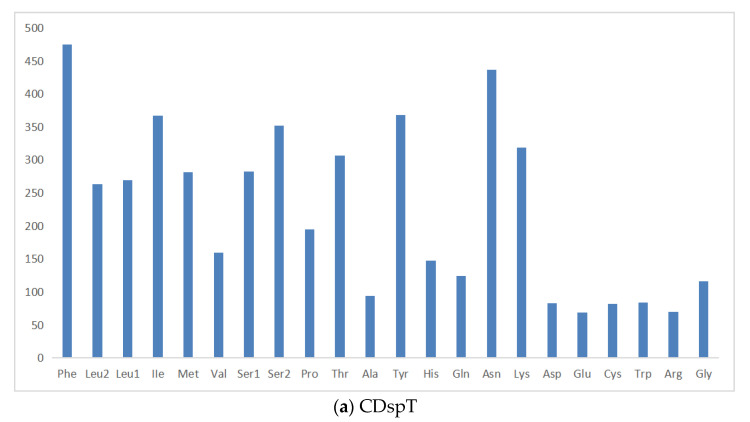
Amino acid composition in *P. makianus* mitogenome. Codons are expressed in thousands and the x-axis indicates the codon families (**a**) and the relative synonymous codon usage (**b**).

**Figure 3 genes-13-01943-f003:**
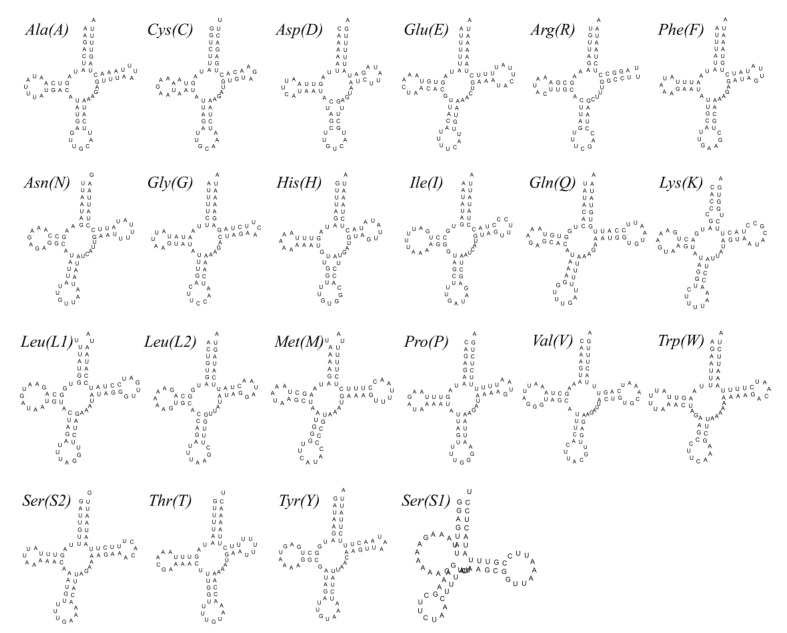
Secondary structures of tRNAs of the *P. makianus* mitogenome.

**Figure 4 genes-13-01943-f004:**
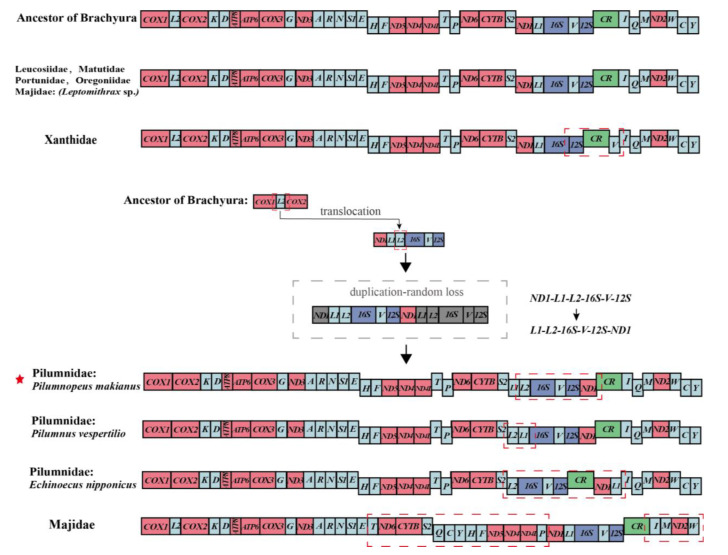
Linear portrayal of gene arrangements of the ancestor Brachyura, *P. makianus*, and six families in Heterotremata. The single-letter amino acid codes represent corresponding tRNA, and the *16S* and *12S* are the large and small ribosomal RNA subunits, respectively. The rearranged gene blocks are signed by red gridlines and compared with the gene arrangement of ancestral Brachyura.

**Figure 5 genes-13-01943-f005:**
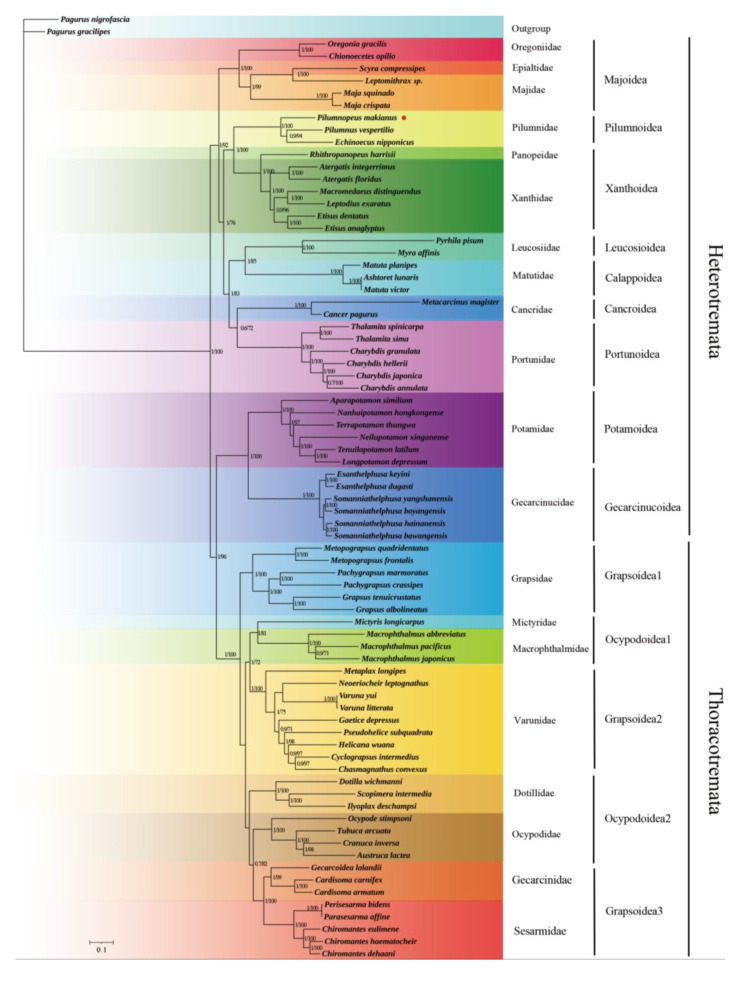
The phylogenetic tree constructed using the 13 PCGs, with BI and ML methods. Numbers above branches are posterior probabilities from BI (left) and bootstrap percentages from ML (right), respectively.

**Table 1 genes-13-01943-t001:** Features of the mitochondrial genome of *P. makianus*.

Gene	Position		length	Amino Acid	Start/Stop Codon	Anticodon	Intergenic Region	Strand
	from	to						
*COX1*	1	1539	1539	513	ATG/TAA	-	49	H
*COX2*	1589	2293	705	235	ATG/TAA	-	−20	H
*Lys*	2274	2340	67	-	-	TTT	5	H
*Asp*	2346	2408	63	-	-	GTC	0	H
*ATP8*	2409	2567	159	53	ATG/TAG	-	59	H
*ATP6*	2627	3235	609	203	ATG/TAA	-	−1	H
*COX3*	3235	4026	792	264	ATG/TAA	-	−1	H
*Gly*	4026	4090	65	-	-	TCC	0	H
*ND3*	4091	4444	354	118	ATG/TAA	-	4	H
*Ala*	4449	4515	67	-	-	TGC	4	H
*Arg*	4520	4583	64	-	-	TCG	0	H
*Asn*	4584	4650	67	-	-	GTT	6	H
*Ser1*	4657	4723	67	-	-	TCT	0	H
*Glu*	4724	4791	68	-	-	TTC	30	H
*His*	4822	4885	64	-	-	GTG	3	L
*Phe*	4889	4952	64	-	-	GAA	1	L
*ND5*	4954	6678	1725	575	ATT/TAA	-	62	L
*ND4*	6739	8070	1332	444	ATG/TAA	-	−5	L
*ND4L*	8064	8366	303	101	ATG/TAA	-	2	L
*Thr*	8369	8432	64	-	-	TGT	0	H
*Pro*	8433	8497	65	-	-	TGG	17	L
*ND6*	8515	9006	492	164	ATT/TAA	-	−1	H
*CYTB*	9006	10,140	1135	378	ATG/T(AA)	-	0	H
*Ser2*	10,141	10,207	67	-	-	TGA	45	H
*Leu1*	10,253	10,322	70	-	-	TAG	23	L
*Leu2*	10,346	10,410	65	-	-	TAA	−56	L
*16S*	10,355	11,709	1355	-	-	-	14	L
*Val*	11,724	11,795	72	-	-	TAC	0	L
*12S*	11,796	12,626	831	-	-	-	99	L
*ND1*	12,724	13,656	933	311	ATA/TAA	-	776	L
*Ile*	14,431	14,498	68	-	-	GAT	10	H
*Gln*	14,509	14,577	69	-	-	TTG	10	L
*Met*	14,588	14,656	69	-	-	CAT	0	H
*ND2*	14,657	15,664	1008	336	ATG/TAA	-	1	H
*Trp*	15,666	15,733	68	-	-	TCA	−1	H
*Cys*	15,733	15,795	63	-	-	GCA	0	L
*Tyr*	15,796	15,861	66	-	-	GTA	2	L

**Table 2 genes-13-01943-t002:** Composition and skewness of *P. makianus* mitogenome.

	A%	T%	G%	C%	(A + T) %	AT-Skew	GC-Skew	Length(bp)
Mitogenome	35.76	36.5	10.32	17.42	72.26	–0.010	–0.256	15,863
PCGs	29.49	40.83	14.95	14.74	70.32	–0.161	0.007	11,066
*cox1*	29.37	35.67	15.98	18.97	65.04	–0.097	–0.086	1539
*cox2*	32.26	37.52	12.12	18.1	69.78	–0.075	–0.198	685
*atp8*	32.08	42.14	7.55	18.24	74.22	–0.136	–0.415	159
*atp6*	29.56	40.07	12.48	17.9	69.63	–0.151	–0.178	609
*cox3*	28.54	38.76	14.9	17.8	67.3	–0.152	–0.089	792
*nad3*	31.64	41.81	11.58	14.97	73.45	–0.138	–0.128	354
*nad5*	30.72	40.75	18.9	9.62	71.47	–0.140	0.325	1725
*nad4*	29.43	45.05	17.34	8.18	74.48	–0.210	0.359	1332
*nad4L*	29.04	43.56	21.78	5.61	72.6	–0.200	0.590	303
*nad6*	27.64	47.15	7.32	17.89	74.79	–0.261	–0.419	492
*cob*	29.43	37.71	13.48	19.38	67.14	–0.123	–0.180	1135
*nad1*	26.47	44.69	19.72	9.11	71.16	–0.256	0.368	933
*nad2*	29.17	43.06	8.13	19.64	72.23	–0.192	–0.414	1008
tRNAs	37.41	35.7	15.53	11.35	73.11	0.023	0.156	1462
rRNAs	39.22	36.94	15.75	8.08	76.16	0.030	0.322	2190

**Table 3 genes-13-01943-t003:** Nucleotide composition of the mitogenomes of three Pilumnidae species.

Species	Length(bp)	A%	T%	G%	C%	(A + T) %	AT-Skew	GC-Skew	
Complete mitogenome									
*P. makianus*	15,863	35.76	36.5	10.32	17.42	72.26	–0.010	–0.256	
*P. vespertilio*	16,222	35.26	35.82	10.45	18.47	71.08	–0.01	–0.28	
*E. nipponicus*	16,173	35.6	35.64	10.23	18.52	71.24	0.00	–0.29	
PCGs									
*P. makianus*	11,066	29.49	40.83	14.95	14.74	70.32	–0.161	0.007	
*P. vespertilio*	11,171	28.81	40.07	15.54	15.58	68.88	–0.16	0.00	
*E. nipponicus*	11,189	28.62	40.26	15.55	15.57	68.88	–0.17	0.00	
tRNAs									
*P. makianus*	1462	37.41	35.7	15.53	11.35	73.11	0.023	0.156	
*P. vespertilio*	1479	37.25	35.97	15.48	11.29	73.22	0.02	0.16	
*E. nipponicus*	1440	38.12	36.53	13.75	11.6	74.65	0.02	0.08	
rRNAs									
*P. makianus*	2190	39.22	36.94	15.75	8.08	76.16	0.030	0.322	
*P. vespertilio*	2222	39.38	36.23	36.23	7.79	75.61	0.04	0.65	
*E. nipponicus*	2144	38.25	37.17	16.74	7.84	75.42	0.01	0.36	

## Data Availability

The mitochondrial genome was deposited at NCBI, with accession number OM461360. The data that support the finding of this study are openly available in Microsoft OneDrive at https://1drv.ms/u/s!Agslj0zkcUG8gRAZPuvOxsXtz3aQ?e=2CxF5k(accessed on 17 January 2022). The raw genome sequencing datasets generated during the current study have been submitted to the NCBI Sequence Read Archive (SRA) and are available online at https://www.ncbi.nlm.nih.gov/bioproject/PRJNA885758 (accessed on 30 September 2022).

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
