# Peer review of "The Complete Mitochondrial Genome of Pilumnopeus Makianus (Brachyura: Pilumnidae), Novel Gene Rearrangements, and Phylogenetic Relationships of Brachyura"

_genes, 2022, doi:10.3390/genes13111943_

Round 1

Reviewer 1 Report

In this study, the authors firstly revealed the complete mitogenome of Pilumnopeus makiana and the specific classification status of the species is further confirmed by molecular genetic characteristics. I enjoyed to reads the manuscript and it is well analyzed and discussion is also well described based on the outcomes. I have a few minor comments for the revision.

1, L83: How many specimens were collected and examined in this study and when did you collect them? Please state the details.

2, L110: It would be better to make a table or add as an appendix for details of specimens, especially location which was downloaded from NCBI. 

3, L274: Can you discuss the significance that your result was consistent with the previous study by Chen et al. [51]? It seems to somewhat uninteresting if the conclusion is just “The same results were obtained in our study.” in the paragraph.

4, L288: You stated “The phylogenetic tree was structured by 77 species”. However, “Pilumnidea is widely distributed along Indo-West Pacific coast, there are species 404 of 68 genera belonging to 3 families” in L44-45. So, if you can study all species, How the tree will be structured? Since you only examined less than 20% in Pilumnidea, if you can discuss on it, it would be useful for further study. 

Author Response

Reviewer1

In this study, the authors firstly revealed the complete mitogenome of Pilumnopeus makiana and the specific classification status of the species is further confirmed by molecular genetic characteristics. I enjoyed to reads the manuscript and it is well analyzed and discussion is also well described based on the outcomes. I have a few minor comments for the revision.

Answer: Thank you for your comments on the manuscript. Efforts have been made to clarify the following points and correct the mistakes. The modifications based on the suggestions are explained in further detail below. We have directly revised some minor changes. The corresponding revised manuscript and annotated version are submitted.

1, L83: How many specimens were collected and examined in this study and when did you collect them? Please state the details.

Answer: Relevant information has been supplemented.

2, L110: It would be better to make a table or add as an appendix for details of specimens, especially location which was downloaded from NCBI.

Answer: We have added the Table S1.

3, L274: Can you discuss the significance that your result was consistent with the previous study by Chen et al. [51]? It seems to somewhat uninteresting if the conclusion is just “The same results were obtained in our study.” in the paragraph.

Answer: the sentence has been rewritten.

4, L288: You stated “The phylogenetic tree was structured by 77 species”. However, “Pilumnidea is widely distributed along Indo-West Pacific coast, there are species 404 of 68 genera belonging to 3 families” in L44-45. So, if you can study all species, How the tree will be structured? Since you only examined less than 20% in Pilumnidea, if you can discuss on it, it would be useful for further study.

Answer: Because the phylogenetic tree was structured by the whole mitochondrial genome, there are only two belonging to Pilumnidea in NCBI, i.e. Pilumnus Vespertilio (MF457402) and Echinoecus nipponicus (NC_039618).

Reviewer 2 Report

Minor editorial revisions will be needed.

1. line 22: Pilumnopeus should be P.

2. line 24: delete will ?

3. lines 46, 47: Pilumnopeus should be P. ?

4. line 88: delete space between 1.5 and %

5. line 92: add (company name, city, state, country) after platform

6. line 100: de novo should be de novo

7. line 129: discussion should be Discussion ?

8. line 130: base should be Base

9. lines 153, 158, 175, 213, 226  Pilumnopeus should be P. ?

10. line 193: I.e. should be i.e. ?

11. lines 232, 240, 282: Piliumnopeus should be P. ?

12. line 268: add et al. after Ahyong

13. line 268: Bracken should be Brosing et al.

14. line 313: A.;L.; Rice should be Rice, A.L.

15. line 325, 333-334, 340-341, 357, 361-362, 363-364, 379-380, 390-391, 407-408: title should  be small capital

16. line 332: res. should be Res.

17. line 351: MITOS should be Mitos

18. line 368: is this a book title or book chapter ?  or journal?

19. line 373: Brassica napus shouls be italic

20. line 370: Region should be region

Author Response

Reviewer 2:

Minor editorial revisions will be needed.

Answer: Thank you for your comments on the manuscript. Efforts have been made to clarify the following points and correct the mistakes. The modifications based on the suggestions are explained in further detail below. We have directly revised some minor changes. The corresponding revised manuscript and annotated version are submitted.

A point-by-point response to the comments

  1. line 22: Pilumnopeus should be P.

Answer: After inspection, we found that Pilumnopeus should be Pilumnus, and the word has been rewritten.

  1. line 24: delete will ?

Answer: “will” has been deleted.

  1. lines 46, 47: Pilumnopeus should be P. ?

Answer: After inspection, we found that Pilumnopeus should be Pilumnus, and the word has been rewritten.

  1. line 88: delete space between 1.5 and %

Answer: “the space between 1.5 and %” has been deleted.

  1. line 92: add (company name, city, state, country) after platform

Answer: “(company name, city, state, country)” has been added.

  1. line 100: de novo should be de novo

Answer: The word has been rewritten.

  1. line 129: discussion should be Discussion ?

Answer: The word has been rewritten.

  1. line 130: base should be Base

Answer: The word has been rewritten.

  1. lines 153, 158, 175, 213, 226 Pilumnopeus should be P. ?

Answer: All the words have been rewritten.

  1. line 193: I.e. should be i.e. ?

Answer: The word has been rewritten.

  1. lines 232, 240, 282: Piliumnopeus should be P. ?

Answer: All the words have been rewritten.

  1. line 268: add et al. after Ahyong

Answer: “et al.” has been added.

  1. line 268: Bracken should be Brosing et al.

Answer: “et al.” has been added.

  1. line 313: A.;L.; Rice should be Rice, A.L.

Answer: The name has been rewritten.

  1. line 325, 333-334, 340-341, 357, 361-362, 363-364, 379-380, 390-391, 407-408: title should be small capital

Answer: All the titles have been rewritten.

  1. line 332: res. should be Res.

Answer: The word has been rewritten.

  1. line 351: MITOS should be Mitos

Answer: The word has been rewritten.

  1. line 368: is this a book title or book chapter ? or journal?

Answer: Relevant information has been added.

  1. line 373: Brassica napus should be italic

Answer: The word has been rewritten.

  1. line 370: Region should be region

Answer: The word has been rewritten.